# Pro-Fibrotic Effects of CCL18 on Human Lung Fibroblasts Are Mediated via CCR6

**DOI:** 10.3390/cells13030238

**Published:** 2024-01-26

**Authors:** Kerstin Höhne, Annett Wagenknecht, Corinna Maier, Peggy Engelhard, Torsten Goldmann, Stephan J. Schließmann, Till Plönes, Martin Trepel, Hermann Eibel, Joachim Müller-Quernheim, Gernot Zissel

**Affiliations:** 1Department of Pneumology, Medical Center–University of Freiburg, Faculty of Medicine, University of Freiburg, 79106 Freiburg, Germany; kerstinhoehne@gmx.de (K.H.); corinna.maier@gmx.net (C.M.); peggy.engelhard@gmx.de (P.E.); stephan.schliessmann@mainsport.info (S.J.S.); joachim.mueller-quernheim@uniklinik-freiburg.de (J.M.-Q.); 2Department of Medicine I, Medical Center–University of Freiburg, Faculty of Medicine, University of Freiburg, 79106 Freiburg, Germany; annett.kilic@googlemail.com (A.W.); martin.trepel@uk-augsburg.de (M.T.); 3Histology, Research Center Borstel, 23845 Borstel, Germany; tgoldmann@fz-borstel.de; 4Integrative and Experimental Exercise Science and Training, Institute of Sport Science, University of Würzburg, 97082 Würzburg, Germany; 5Department of Thoracic Surgery, Center for Surgery, Medical Center–University of Freiburg, Faculty of Medicine, University of Freiburg, 79106 Freiburg, Germany; till.ploenes@ukdd.de; 6Department of Internal Medicine II, University Medical Center and Medical Faculty, Augsburg University, Germany Internal Medicine and Oncology, Faculty of Medicine, University of Augsburg, 86156 Augsburg, Germany; 7Center for Chronic Immunodeficiency, Medical Center–University of Freiburg, Faculty of Medicine, University of Freiburg, 79106 Freiburg, Germany; hermann.eibel@uniklinik-freiburg.de

**Keywords:** idiopathic pulmonary fibrosis, fibrogenesis, chemokine receptor, ligand-induced internalization, co-immunoprecipitation, collagen, alpha-smooth muscle actin, FGF2

## Abstract

Background: Idiopathic pulmonary fibrosis (IPF) is a fatal lung disease of unknown origin, with a median patient survival time of ~3 years after diagnosis without anti-fibrotic therapy. It is characterized by progressive fibrosis indicated by increased collagen deposition and high numbers of fibroblasts in the lung. It has been demonstrated that CCL18 induces collagen and αSMA synthesis in fibroblasts. We aimed to identify the CCL18 receptor responsible for its pro-fibrotic activities. Methods: We used a random phage display library to screen for potential CCL18-binding peptides, demonstrated its expression in human lungs and fibroblast lines by PCR and immunostaining and verified its function in cell lines. Results: We identified CCR6 (CD196) as a CCL18 receptor and found its expression in fibrotic lung tissue and lung fibroblast lines derived from fibrotic lungs, but it was almost absent in control lines and tissue. CCL18 induced receptor internalization in a CCR6-overexpressing cell line. CCR6 blockade in primary human lung fibroblasts reduced CCL18-induced FGF2 release as well as collagen-1 and αSMA expression. Knockdown of CCR6 in a mouse fibroblast cell line abolished the induction of collagen and α-smooth muscle actin expression. Conclusion: Our data indicate that CCL18 triggers pro-fibrotic processes via CCR6, highlighting its role in fibrogenesis.

## 1. Introduction

Fibrogenesis is a key pathomechanism in various pulmonary diseases, particularly idiopathic pulmonary fibrosis (IPF) [1,2,3] and non-specific interstitial pneumonia (NSIP), but also occurs in end-stage disease lungs of patients with hypersensitivity pneumonitis (HP) or sarcoidosis (SAR). Fibrogenesis leads to a remodelling of the delicate lung micro-architecture into compact and rigid connective tissue. Due to the severe loss of lung function and the lack of an effective therapeutic option in IPF [4], the mean patient survival time ranges between 2.5 and 3.5 years after diagnosis [2,5,6], but some improvement has been achieved with newly approved therapeutics like nintedanib and pirfenidone, which target signalling in the FGF, PDGF and TGFβ pathways [7,8,9]. The prognosis of non-IPF pulmonary fibrosis is in most cases more favourable compared with IPF, although no approved therapeutics exist [10,11].

Current hypotheses on the onset of fibrosis postulate epithelial damage with ongoing repair processes either due to recurrent inflammation or repeated injuries. Thus, fibrosis is regarded as a result of an abnormal wound-healing process in which aberrant cross-talk between fibroblasts and epithelial cells promotes chronic fibroblast proliferation [12,13,14,15]. This causes the release of several growth factors and cytokines resulting in an increased migration and proliferation of fibroblasts. Thus, fibroblasts are the key players in pulmonary fibrosis, and their enhanced extracellular matrix deposition is a hallmark of pulmonary fibrosis [12,16].

Therapeutic options are rare, as anti-inflammatory therapeutic approaches like steroids or cyclophosphamide have only little impact on fibrotic processes of the lung. While these drugs are of value in distinct forms of pulmonary fibrosis characterized by an inflammatory cell composition, e.g., non-specific-interstitial pneumonia (NSIP) [6,17], they are hardly effective for IPF, which is the most common form of pulmonary fibrosis. Currently, two new drugs have been approved for the therapy of mild to moderate fibrosis in IPF; however, although these drugs slow down disease progression, they do not induce a “restitutio ad integrum” [7,8].

CC-chemokine 18 (CCL18) is a chemokine with chemotactic and immuno-regulatory functions mainly released by alternatively activated macrophages [18,19,20,21,22]. Interestingly, CCL18 is unique in the chemokine system, as it is present only in humans, and there is currently no comparable chemokine known in rodents [23,24]. We found elevated CCL18 serum levels and increased CCL18 release by alveolar macrophages from patients suffering from pulmonary fibrosis [25]. Moreover, we have demonstrated that both CCL18 production by bronchoalveolar lavage (BAL) cells and serum CCL18 concentrations reflect pulmonary fibrotic activity [26] and are prognostic markers of IPF [27,28,29].

The importance of CCL18 in fibrotic diseases is obvious by its collagen-inducing properties [30]. High levels of CCL18 are directly linked to the increased matrix deposition in pulmonary fibrosis [25]. Vice versa, we have demonstrated that collagen recognition via CD204 increases CCL18 release by alveolar macrophages [31] inducing a vicious cycle in IPF [25]. Although the pro-fibrotic properties of CCL18 are already known, the transmission of the CCL18 signals are not clear because, for a long time, CCL18 was one of the few chemokines with unknown receptor(s). We therefore aimed to identify the receptor for CCL18 responsible for its pro-fibrotic properties. The identification of such a receptor would provide a new therapeutic target in disorders with increased levels of CCL18 like fibrosis [25] and cancer [32]. Our data suggest that the CC-chemokine receptor 6 (CCR6) expressed on fibroblasts mediates the pro-fibrotic activity of CCL18.

## 2. Materials and Methods

### 2.1. Phage Display Peptide Library Screenings

A cyclic random phage display peptide library with seven random residues (CX_7_C; C = cysteine; X = random amino acid) was cloned and generated as described previously for phage [33] and for plasmid libraries in the context of the adeno-associated viral genome [34]. The library was screened based on recombinant CCL18 for epitope mapping after two-fold negative selection based on irrelevant control proteins BSA and casein. The screening was performed for three selection rounds. Randomly selected clones from the third panning round were sequenced (GATC, Konstanz, Germany).

### 2.2. Patients

Fibroblast lines were established either from surgical material, explanted lungs (n = 13), pneumectomies or lobectomies of patients suffering from squamous carcinoma (n = 19) or from left-over material from diagnostic biopsies of fibrotic lungs (video-assisted thoracoscopy (VATS, n = 4) or transbronchial biopsies (TBB, n = 4) of patients suffering from idiopathic pulmonary fibrosis (IPF, n = 13), non-specific interstitial pneumonitis (NSIP, n = 3), sarcoidosis, (n = 3) or hypersensitivity pneumonitis (HP, n = 2)). All cases of NSIP were “non-classified NSIP”, as no underlying diseases could be detected.

### 2.3. Immunohistochemistry

Tumour-free lung tissues were obtained from surgery (lobectomy or pneumonectomy) for non-small cell lung cancer; only specimens at least 7 cm away from the tumour were selected as tumour-free, and only those confirmed by microscopy were entered into the study. Tissues were fixed and embedded in paraffin using the HOPE technique [35]. Immunohistochemical detection of CC-chemokine receptor 6 (CCR6) was performed as previously described [35,36]. Primary antibody (anti-hCCR-6 mouse monoclonal IgG, clone 53,103, 500 µg/mL, R&D Systems, Minneapolis, MN, USA) was applied after de-paraffinization in a dilution of 1/100 for 1 h at ambient temperature. Detection of signals was achieved with the ZytoChem Plus HRP polymer system (Zytomed Systems, Berlin, Germany) according to the instructions of the manufacturer. Aminoethylcarbazole (permanent AEC, Zytomed Systems, Berlin, Germany) was used as a chromogen.

### 2.4. Isolation of Primary Fibroblasts

Lung tissue was cut in small pieces (app. 0.5 cm edge length) and placed in 6-well plates containing 1 mL Quantum 333 (PAA, Pasching, Austria) with 1% penicillin/streptomycin. Quantum 333 is a fibroblast selection medium that allows the growth of cells without the addition of foetal calf serum. Outgrowing fibroblasts were harvested by trypsinisation when they reached approximately 80% confluence, cultured in 75 cm^2^ cell culture flasks (NUNC Thermo Fisher, Roskilde, Denmark) in Quantum 333 or DMEM with 10% FCS and subsequently split (1:3 to 1:5). The established lines were enrolled in our studies from passage 3, to ensure that no other cells (e.g., macrophages) were in our culture, to passage 8, to prevent senescence of the cells. Phase-contrast microscopy photographs of three different cell lines are depicted in Appendix A.

### 2.5. Cell Lines

Rat lung epithelial cells were obtained from ATCC (Teddington, Middlesex, UK) and cultured in HAM-F12 (Lonza, Verviers, Belgium) containing 10% foetal bovine serum (FCS gold, PAA, Pasching, Austria), 10 µg/mL bovine pituitary extract (Gibco, Invitrogen, Paisley, UK), 5 µg/mL insulin (Sigma, Taufkirchen, Germany), 2.5 ng/mL insulin-like growth factor (ImmunoTools, Friesoythe, Germany), 1.25 µg/mL transferrin (Gibco, Invitrogen, Paisley, UK) and 2.5 ng/mL EGF (ImmunoTools, Friesoythe, Germany).

U937 cells are a pro-monocytic lymphoma cell line originally derived from a male patient. We found that in our laboratory cell line, a large percentage of cells express CCR6. We therefore used anti-CCR6 antibodies (R&D Systems, Minneapolis, MN, USA) and anti-mouse IgG microbeads (Miltenyi, Bergisch-Gladbach, Germany) and MACS cell enrichment columns (Miltenyi) to enrich CCR6-positive U937 cells in order to establish a “U937*CCR6+” cell line (Appendix A). The cells were cultured in DMEM (Gibco, Invitrogen, Paisley, UK) containing 10% foetal bovine serum (FCS gold, PAA, Pasching, Austria) and 1% penicillin/streptomycin (Gibco, Invitrogen, Paisley, UK).

The mouse embryonal fibroblast line NIH/3T3 was obtained from ATCC and cultured in DMEM (Gibco, Invitrogen, Paisley, UK) containing 10% foetal bovine serum (FCS gold, PAA, Pasching, Austria) and 1% penicillin/streptomycin (Gibco, Invitrogen, Paisley, UK). All cells were incubated at 37 °C and 5% CO_2_.

### 2.6. Flow Cytometry

Fibroblast populations were gated by forward/sideward scatter, and surface expression of CCR6 was estimated using a FITC-labelled anti-human CCR6 antibody (R&D Systems, Minneapolis, MN, USA) and a suitable IgG control. As CCR6 is a trypsin-sensitive molecule, trypsinized cells were allowed to recover in medium in polypropylene tubes (to avoid adhesion) for at least 3 h but not more than 4 h at 37 °C before staining.

### 2.7. FGF2 Determination

Fibroblasts were seeded at a density of 300.000 cells per well in a 6-well cell culture plate in Quantum 333 medium. Cells were allowed to attach overnight, and the medium was replaced with 1 mL of DMEM plus 10% FCS. Cells were either left non-stimulated or were stimulated with 10 ng/mL of CCL18. To block CCR6/CCL18 interaction, a blocking antibody against CCR6 (R&D systems, Wiesbaden, Germany, FRG) or an irrelevant antibody (mouse anti human IgG, R&D Systems) was added to a final concentration of 20 µg/mL in parallel cultures. After 24 h of culture, the supernatant was harvested and stored at −80 °C until FGF2 determination. FGF2 was measured using an ELISA development kit (R&D) and performed as suggested by the supplier.

### 2.8. Real-Time PCR

After the indicated culture period, total RNA was extracted using TRIzol Reagent according to the manufacturer’s protocol (Invitrogen, Karlsruhe, Germany). Total RNA was reverse-transcribed with an iScript cDNA Synthesis Kit (BIO-RAD, Hercules, CA, USA) using oligo (dT) and random primers to produce cDNA according to the manufacturer’s protocol. Specific primers for human CCR6, collagen 1, alpha-smooth muscle actin and GAPDH were designed using AmplifX 1.7.0 (https://inp.univ-amu.fr/en/amplifx-manage-test-and-design-your-primers-for-pcr) [37] using the GenBank database (National Center for Biotechnology Information; https://www.ncbi.nlm.nih.gov/genbank/). Accession code numbers for the nucleotide sequences used to generate the respective primers and the primer sequences are depicted in Table 1.

All primers were intron-spanning and synthesized with biomers.net (Ulm, Germany). Real-time PCR was performed with the iQ SYBR Green SuperMix, iCycler thermocycler and iCycler iQ 3.0 software (Bio-Rad Laboratories GmbH, Feldkirchen, Germany) according to the manufacturer’s protocol. After initial denaturation (10 min, 95 °C), PCR was performed using an annealing time of 30 s at 57 °C, amplification for 30 s, and 15 s at 94 °C for denaturation for 45 cycles. To control for specificity of the amplification products, a melting curve analysis was performed. No amplification of nonspecific products was observed in any of the reactions. A threshold cycle value (Ct) was calculated and used to compute the relative level of specific mRNA by the following formula:“relative expression = 2^(Ct GAPDH−Ct(target))^ × 10,000” for each cDNA sample

### 2.9. Stable Transfection of RLE-6TN

As the human CCR6 vector (pORF9-hCCR6, InvivoGen, San Diego, CA, USA) does not have a Sal I or Sma I cleavage site, the vector was first cut with EcoRV, and a Sal I linker was integrated into the cleavage site. In order to create a GFP fusion protein, a part of the coding sequence of CCR6 was re-synthesized together with a Sma I cleavage site, and the stop codon at the C-terminal end was removed and ligated back into the original vector. Only then was the cleavage carried out with Sal I and Sma I.

In total, 3–5 × 10^5^ 293T cells were transfected with pCDNL B**, pLTR VSVG and pNL CEV CCR6*GFP using FuGeneHD (Roche Diagnostics, Penzberg, Germany). After 24 h, the medium was replaced, and after 48 h, the viral supernatant was used to infect RLE-6TN cells.

Stable transfected RLE-CCR6*GFP cells were separated from wild-type cells by fluorescence-activated cell sorting and limiting dilution. The purity of the cells was 99% and was stable during the culture periods.

### 2.10. Knock Down of CCR6 in NIH/3T3 Cells by CRISPR/Cas9

Primer sequences for the generation of CRISPR/Cas9 were selected with the gRNA design tool by ThermoFischer (ThermoFisher, Waltham, MA, USA) (thermofisher.com/crisprdesign) and synthesized with biomers.net. The following primer sequences were used: ATAATCATCCGTTCCAAAGT and GTAGACGTCAGTCATGGATC. The gRNA was then synthesized and transcribed using the GeneArt Precision gRNA Synthesis Kit (Invitrogen/ThermoFisher, Waltham, MA, USA). GeneArt Platinum Cas9 Nuclease protein (Invitrogen/ThermoFisher) and the gRNAs were then used to transfect the cells using the Lipofectamine™ CRISPRMAX™ Transfection Reagent (Invitrogen/ThermoFisher) according to the protocol provided by the manufacturer. After transfection, the cells were cultured for two days, harvested and cloned by limiting dilution. Clones were picked, expanded and analysed for CCR6 expression. CCR6-negative clones were expanded and used for the experiments.

### 2.11. Receptor Internalization

Receptor internalization was estimated using flow cytometry and fluorescence microscopy. CCR6 is a trypsin-sensitive receptor; thus, it is lost during the detachment of adherent cells using trypsin. To avoid artefacts by damaging CCR6 by trypsinization, we used the non-adherent U967*CCR6+ cell line. The cells were incubated with CCL18 or CCL20 in varying concentrations ranging from 0.03 to 1000 nM in 30 µL of culture medium containing 10,000 cells for 20 min and immediately fixed by addition of 200 µL of cold paraformaldehyde (4%, Sigma, Taufkirchen, Germany) for 5 min on ice. The cells were then washed by the addition of 1 mL PBS and centrifugation. This step was performed twice. Subsequently, the cells were stained with a FITC-labelled anti-human CCR6 antibody (R&D Systems, Minneapolis, MN, USA) and analysed using a FACS Calibur (Becton Dickinson, Heidelberg, Germany) system and FlowJo software (version 8.5.3) for data analysis (Becton Dickinson, Heidelberg, Germany).

For fluorescence microscopy, RLE-CCR6*GFP cells were seeded on chamber slides and cultured overnight at 37 °C and 5% CO_2_ to allow adherence. The next day, the medium was discarded and replaced with medium containing 10 ng/mL of CCL18. As a control, we added CXCL10 in the same concentration. After the indicated time points, the cells were fixed and embedded in Vecta shield containing DAPI (Vector Laboratories, Newark, CA, USA) and analysed using a fluorescence microscope (Olympus, Hamburg, Germany).

### 2.12. Co-Immunoprecipitation

For co-immunoprecipitation (Co-IP), 107 RLE-6TN CCR6*GFP cells were cultured overnight in a petri dish to let them adhere; they were subsequently loaded with 500 ng/mL recombinant human CCL18 (Immunotools, Friesoyte, Germany) for 10 and 30 min. After incubation, loaded cells were gently lysed using the lysing buffer provided by the manufacturer, and CCR6*GFP was isolated using the isolation kit for GFP-tagged proteins according to the manufacturer’s protocol (Miltenyi, Bergisch Gladbach, Germany). The isolated CCR6*GFP and CCR6*GFP/CCL18 complexes were analysed by routine western blotting. In brief, isolated proteins were boiled at 95 °C for 5 min in equal volumes of loading buffer (0.5 M Tris-HCl pH 6.8, 2% SDS, 0.05% bromophenol blue, 20% 2-mercaptoethanol and 10% Glycerol) and subjected to 12% sodium dodecyl sulphate–PAGE, separated by electrophoresis and transferred to a polyvinylidene difluoride membrane (PVDF). After blocking for 2 h in Tris-buffered saline (TBS) containing 5% non-fat dry milk, the membranes were incubated with primary antibody against CCR6 (CKR6/C20, Santa Cruz, California) and anti-CCL18 (R&D Systems, Minneapolis, MN, USA) both diluted 1:700 with TBS at 4 °C overnight. Visualization was performed using appropriate secondary antibodies labelled with IRDye 800 CW or IRDye 700 CW (Li-COR Bioscience, Bad Homburg, Germany) diluted 1:10,000–1:20,000 for 2 h and scanned using an Odyssey system (Li-COR Bioscience) according to the manufacturer’s instructions.

### 2.13. Statistical Analysis

The data are shown as single dots with median or bar charts. As the values of our data do not follow a Gaussian distribution (tested with the Kolmogorow–Smirnov test with Lilliefors correction), we used the non-parametric Mann–Whitney test for comparison of independent samples and the Wilcoxon signed-rank test for connected data. In cases of multiple testing, the data were first tested for general significant differences using Kruskal–Wallis (independent data) or Friedman statistics (matched data). In the single comparisons of the individual groups, the values of *p* were adjusted by Bonferroni–Holm correction using a web-based calculator [38]. All other tests were performed using StatView for Windows (SAS Institute, Cary, NC, USA). Values of *p* ≤ 0.05 were considered statistically significant.

## 3. Results

### 3.1. Identification of a CCL18-Binding Peptide Using Phage Display

To identify peptides binding to CCL18 by phage display, we used a random CX7C phage peptide library (C indicates cysteine; X indicates any amino acid) and selected the binding phages on immobilized CCL18. Competent *E. coli* cells were added to the bound phages for uptake and expansion. Phages were specifically enriched on CCL18 after three rounds of selection. Fifteen clones of the phages derived from the selection on CCL18 were sequenced, and two of them were found to display variants of a CCR6 motif (CTCGCGACTGGTGTGGTGTTT (AL121935) and TGGTGAGCTGGAGTCATCAGA (NM_031409)).

### 3.2. CCR6 Expression in Human Fibrotic Lung Tissue

To identify CCR6 expression in human lung tissue samples, we performed immune-histochemical studies on tissue sections taken from lungs of patients with IPF and lung tissue from lobectomies far from the tumour. Analysis of lung tissue from IPF patients revealed intense CCR6 expression in bronchial epithelia (overview in Figure 1A), hyperplastic type II alveolar epithelial cells (Figure 1B), fibroblasts (Figure 1C) and macrophages (Figure 1D). In contrast, in tumour-free lung tissue samples from a bronchial carcinoma patient (Figure 1E), no CCR6 expression could be detected.

Fibroblasts are regarded as important drivers of fibrotic lung diseases [39], and CCL18 has been reported to foster collagen and αSMA expression in fibroblasts [30,40]. Thus, although we found CCR6 expression on several cell types within our fibrotic lung sections, we concentrate here on the role of the CCR6/CCL18 interaction in fibroblasts.

### 3.3. CCR6 Expression in Human Primary Fibroblast Lines

Next, we determined the mRNA levels of CCR6 in cultured human primary fibroblast lines isolated from the lungs of patients with varying pulmonary diseases. The highest expression was found in fibroblast lines derived from the lungs of patients suffering from IPF and non-specific interstitial pneumonia (NSIP) (Figure 2A). Three out of four IPF fibroblast lines and two of three NSIP lines expressed increased levels of CCR6 mRNA. The median was a 3-fold increase; at the extremes, we found a 10–20-fold increase. Fibroblast lines derived from the lungs of patients suffering from squamous carcinoma expressed only background levels of CCR6 mRNA (Figure 2A).

Figure 2B shows the strategy of our flow cytometric analysis of the fibroblast lines. The analysis revealed detectable CCR6 expression on the surface of fibroblast lines derived from patients suffering from NSIP (92 ± 6% positive cells) and IPF (53 ± 33% positive cells, Figure 2C). In only two out of 13 IPF lines, no CCR6 surface expression (<5% positive cells) could be detected. Of note, fibroblast lines established from lungs of patients suffering from end-stage sarcoidosis or hypersensitivity pneumonitis with fibrotic lesions different from IPF did not show relevant CCR6 expression (Figure 2C). CCR6 expression in control fibroblast lines was detected on a very low percentage of cells (<8%; Figure 2C). Remarkably, increased CCR6 expression of fibroblasts from fibrotic lungs remained increased throughout all cell culture passages (tested up to passage 12) and even after freezing and re-culture (Appendix A).

Interestingly, splitting the results according to the sampling method revealed that fibroblast lines grown from TBB expressed the lowest levels of CCR6, lines established from explant lungs had the highest CCR6 expression and lines established from VATS samples exhibited an intermediate expression level. However, none of these differences reached statistical significance most likely due to the low number of samples (Appendix A).

Expression of other putative CCL18 receptors like CCR8 [41] or PITPNM3 [42] was not detected. GPER [43] was either not or only marginally detected (<10% of the cells).

### 3.4. CCL18 Binding to CCR6

To verify that CCR6 is indeed a receptor for CCL18, we conducted several experiments to demonstrate receptor binding and the functional interaction of CCL18 and CCR6.

#### 3.4.1. CCL18 Binds to CCR6 (Co-IP)

In order to verify binding of CCL18 to CCR6, we co-precipitated CCL18 bound to CCR6 by isolation of the CCL18/CCR6*GFP complex using anti-GFP beads. RLE-CCR6*GFP cells were loaded with CCL18, and the CCL18/CCR6*GFP complex was gently isolated from the lysed cells using anti-GFP beads and analysed by Western Blot. Anti-CCR6 antibodies detected CCR6 in both CCL18-loaded and CCL18-non-loaded cells (Figure 3A, lanes 2, 3 and 4: red bands). In contrast, CCL18 was detected when added directly (Figure 3A, lane 1) or in CCL18-loaded cells after 30 min of incubation (Figure 3A, lane 4 bottom: green band).

#### 3.4.2. CCL18-Induced Internalization of CCR6

The RLE-CCR6*GFP cells disclosed strong CCR6 expression as well as bright-green fluorescence (Appendix A). In fluorescence microscopy, the CCR6*GFP complex appears as a clear membrane-associated expression pattern of CCR6 (Figure 3B, upper panel). In addition, there is a light intracellular distribution of CCR6*GFP visible intracellularly. This corresponds to an increase in CCR6 mean fluorescence intensity when we add a membrane-permeabilization step to the CCR6-staining procedure for fibroblasts (Appendix A).

Stimulation of RLE-CCR6*GFP cells with CCL18 (10 ng/mL) induced a time-dependent internalization of the CCR6*GFP complex (Figure 3B, mid panels). After 5 min, the CCR6*GFP complex was seen as green dots or as a ring intracellularly. After 20 min, the CCR6*GFP complex was distributed diffusely within the cells. This internalization demonstrates that CCL18 triggers CCR6. In contrast, an irrelevant chemokine (CXCL10) did not induce CCR6*GFP internalization (Figure 3B, lower panel).

Flow cytometric analysis of CCR6 expression using U937*CCR6^+^ cells revealed a dose-dependent decrease in CCR6 expression with increasing concentrations of CCL18 (Figure 3C). As CCL20 was for a long time the only known chemokine ligand for CCR6, we compared the CCR6 internalization induced by these two chemokines. Of interest, a 10-fold higher concentration of CCL18 was required to reach the same amount of receptor internalization compared with CCL20 (Figure 3B).

### 3.5. Functional Analysis of CCR6

#### 3.5.1. CCL18-Induced Activation of Human Primary Lung Fibroblasts

Non-stimulated fibroblasts derived from lungs of controls released only low levels of FGF2. CCL18 nonsignificantly stimulated FGF2 release in these cells (Figure 4A, dots). In contrast, fibroblast lines derived from fibrotic lungs released increased levels of FGF2; however, these differences did not reach statistical significance. Stimulation with CCL18 significantly increased FGF2 release (Figure 4A, triangles). CCL18-stimulated fibroblasts from fibrosis patients released significantly more FGF2 compared with CCL18-stimulated control fibroblasts.

Blockade of CCR6 by a blocking antibody exerted no effect on CCL18-stimulated FGF2 release by non-fibrotic fibroblast lines. In contrast, in fibroblast lines derived from fibrotic lungs, blockade of CCR6 with a blocking antibody abrogated CCL18-induced FGF2 up-regulation almost completely. The significant difference between CCL18-stimulated fibrosis and control fibroblasts is now lost. In contrast, using an irrelevant IgG isotype control antibody, the CCR6 blockade in fibroblasts from fibrotic patients is incomplete and the difference between these fibroblasts and the control fibroblasts remains significant (Figure 4A).

As reported [25,30], CCL18 induces the expression of collagen and αSMA. We, therefore, determined whether the induction of these molecules is also CCR6-dependent. As shown in Figure 4B (upper panel), CCL18 up-regulated collagen 1 expression in three of four fibrotic-fibroblast lines analysed. This induction was inhibited in all three CCL18 reactive fibroblast lines by the blockade of CCR6 using an inhibitory antibody. The same result was obtained for the CCL18-induced up-regulation of αSMA. Most interestingly, the same cell line that failed to increase collagen 1 expression by CCL18 stimulation also failed to up-regulate αSMA (Figure 4B, lower panel).

#### 3.5.2. CCL18-Induced Activation of Wild-Type and CCR6-KO NIH 3T3 Cells

Approximately 50% of the mouse fibroblast line NIH 3T3 expressed CCR6 (Figure 5, upper panel), whereas the KO clone disclosed no difference to the control antibody indicating that there was no CCR6 expression (Figure 5, lower panel).

We would like to mention that GPER was not expressed on the CCR6-positive clone but was present on the CCR6-KO clone (Appendix A). CCR8 was highly expressed on both clones (Appendix A). PITPNM3 could not be detected on the clones used here.

In wild-type cells, CCL18 induced a dose-dependent increase in collagen 1A (Col1A, Figure 6A, light grey bars) as well as α-smooth muscle actin (αSMA, Figure 6B, light grey bars). TGFβ (5 ng/mL) as well increased the expression of both molecules. In CCR6-KO cells, CCL18 did not induce Col1A expression (Figure 6A, dark grey bars), whereas TGFβ induced a clear-cut up-regulation of Col1A. Interestingly, αSMA is even dose-dependently down-regulated when the CCR6-KO cells are stimulated with CCL18 (Figure 6B, dark grey bars). In contrast, TGFβ induced an up-regulation of αSMA in wild-type cells as well as in the CCR6-KO cells.

## 4. Discussion

CCL18 has been identified as a structural homolog of MIP1α (CCL3) and is primarily released by alternatively activated macrophages [44]. Just three years after this discovery, it was found that CCL18 mRNA is up-regulated in the lungs of patients with hypersensitivity pneumonitis [45]. CCL18 gained interest, as it stimulates collagen production in lung fibroblasts [30,40]. We demonstrated that CCL18 and collagen production are strongly connected [25,31] and that CCL18 is of prognostic usefulness in IPF [26,28]. However, although these results point to an important role of CCL18 in lung fibrosis, no information is available on the receptor responsible for its pro-fibrotic properties. Here, we demonstrate that the pro-fibrotic effects of CCL18 are based on the expression of and the signal transduction by CCR6 in lung fibroblasts.

The role of CCR6 as a CCL18 receptor in fibrotic lung diseases was demonstrated by different approaches. Immunohistochemistry revealed CCR6 expression in human lung tissue from patients suffering from fibrotic lung diseases but not in tissue from lung cancer patients. The data show that different cells are affected including fibroblasts as well as type II alveolar cells or macrophages. Likewise, CCR6 was only expressed on fibroblasts from lungs of patients suffering from fibrotic lung diseases. In contrast, lung fibroblasts from lungs of controls (i.e., tumour-free tissue from cancer patients) did not express CCR6. Thus, CCR6 expression in lung tissue might be a marker for an ongoing fibrotic process. In the current manuscript, we focus on fibroblasts, as these cells are the main drivers of pulmonary fibrosis by their capability to release a large amount of matrix protein. In addition, expression of α-smooth muscle protein turns the fibroblasts into contractible so-called myofibroblasts, a cell type frequently found in pulmonary fibrosis.

CCR6 was first described as a receptor for the chemokine CCL20 [46]. Our data demonstrate that indeed both CCL18 and CCL20 down-regulate CCR6 receptor expression in a dose-dependent manner. However, the concentration required to reach a 50% down-regulation is 10-fold higher for CCL18 compared with CCL20. It has been hypothesized that the binding of CCL20 is dependent on a “DCCL” motif present in the receptor binding region of CCL20 [47]. With a sequence of “LCCL” in this motif is different in CCL18, which might result in weaker binding of CCL18 to CCR6.

Our data point to a constitutive activation of the FGF2 release by the fibroblast lines derived from fibrotic lungs as six out of seven lines released a steady-state level of FGF2 higher than the median of the control group; four out of seven lines released more than the median plus range of the controls (150 ng/mL). These lines were generated just by growth from small lung tissue and expanded over weeks and months. Thus, this increased release cannot be the reverberation of a former in vivo activation. It might be the expression of a phenotypical change in these lung fibroblasts. In addition, in fibroblast lines derived from fibrotic lungs, CCL18 increases FGF2 release as well as collagen and αSMA expression, which is inhibited by anti-CCR6 antibodies. This is of interest, as current drugs like pirfenidone or nintedanib inhibit either the up-regulated release of FGF2 [48] or inhibit signal transduction of the FGF2 receptor [9]. As FGF2 release is up-regulated by CCL18, it is tempting to speculate whether a therapy targeting CCL18/CCR6 interaction is of additive value.

Similarly to FGF2 release, CCR6-dependency of Col1A or αSMA up-regulation was demonstrated by the fact that primary human fibroblasts without CCR6 did not react to CCL18, whereas CCR6-positive fibroblast lines up-regulate Col1A and αSMA after CCL18 stimulation. Again, blockade of CCR6 inhibited this up-regulation of both molecules.

Likewise, NIH 3T3 cells expressing high levels of CCR6 are comparable to human CCR6-positive fibroblast lines, as they show a clear and dose-dependent up-regulation of Col1A and αSMA after CCL18 stimulation. In contrast, up-regulation of both markers is lost after knockdown of CCR6 on these cells. However, stimulation of the CCR6-KO cells with TGFβ induced a robust increase in αSMA and Col1A expression indicating that despite the CCR6 knockdown, these cells are still able to increase αSMA and Col1A using other stimuli. These results strongly suggest that CCR6 expression on fibroblasts is crucial for the pro-fibrotic effects of CCL18.

Most interestingly, a mouse model of bleomycin-induced fibrosis showed that the administration of human CCL18 via an adenoviral vector increased TNF-, IFNγ-, MMP-2 and MMP-9 expression and increased lymphocytosis but attenuated unexpectedly the bleomycin-induced collagen deposition [49]. However, it is not clear whether CCR6 is expressed in the lungs of bleomycin-treated mice. In addition, bleomycin-induced fibrosis is based on a strong inflammatory reaction induced by the drug. CCL18 is known to attract T cells [50,51] and even changes the activity of the recruited T cells [52] and dendritic cells [53]. Thus, it is not clear how a human cytokine, which does not exist in mice, interferes with the mouse immune system in a mouse model of a human disease.

CCR6 is known to be expressed by CD4+ and CD8+ T-cells, by TNF-activated granulocytes, immature dendritic cells and by B cells and human intestinal cells but neither by macrophages nor by freshly isolated monocytes [summarized in [54]]. In contrast, activated macrophages like in active sarcoidosis are highly CCR6-positive [55]. CCR6 gained interest because it was found to be highly expressed on IL-17-producing cells [56] and on FoxP3-expressing IL-17-releasing regulatory T cells [57]. Thus, the expression on fibroblasts and alveolar epithelial cells is remarkable. The reason for this expression currently is unknown; however, it has been reported that in human umbilical vein vascular endothelial cells (HUVEC), CCR6 expression is up-regulated by the combined stimulation with HGF and VEGF [58]. In fibroblast-like synovial cells, stimulation with IL-1β induced an up-regulation of CCR6 expression [59]. In our experiments, up-regulation of CCR6 in normal human lung fibroblasts using IL-4, IL-10, IL-13 and TGFβ in various combinations failed (Appendix A). It is noteworthy that the increased CCR6 expression found in fibroblasts from patients suffering from IPF is stable over several culture passages of the fibroblast lines. Steinfelder et al. showed that during development of memory cells CCR6 expression is epigenetically modulated [60]. Immature DC down-regulate CCR6 expression during maturation [61]. These examples demonstrate that also in other cell types modulation of CCR6 expression is rather an event related to differentiation than to activation. Therefore, we conclude that likewise FGF2 release increased expression of CCR6 by fibroblast lines is not transiently induced by the surrounding cytokine milieu in the fibrotic lung but seems to be a key characteristic of fibroblasts from IPF lungs, which is acquired during pathomechanistic differentiation.

We found a concordant expression of CCR6 on lung fibroblasts demonstrated by immunohistochemistry of fibrotic lung and by flow cytometric analysis of fibroblast lines isolated from lungs of patients suffering from IPF and NSIP. The fact that not all fibroblast lines from IPF lungs express CCR6 might be caused by the heterogeneity of fibroblasts in fibrotic lungs as described earlier [62]. In addition, lung remodelling in IPF is not homogeneously distributed. Severely destructed areas with massive accumulation of fibroblasts and extracellular matrix build a patchwork with areas of almost normal or even completely normal lung histology [63]. In addition, evaluation of our fibroblast lines by the sampling source (explant, VATS, TBB) revealed a clear but nonsignificant difference between the different sampling methods. Explanted tissue was derived from patients with a widely remodelled and dysfunctional end-stage lung. In contrast, VATS and TBB are diagnostic procedures and performed for less advanced lung remodelling. Thus, it is plausible that fibroblasts although grown from an IPF lung sample might be CCR6-negative if this sample was derived from a region of more or less normal histology. However, high CCR6 expression might also be a feature of advanced disease reflecting the tremendous remodelling of the lung.

The lung tissue from patients with sarcoidosis and HP was derived from explanted lungs after lung transplantation due to massive pulmonary fibrosis. Both diseases are based on an inflammatory background and are different from IPF/UIP [10]. Despite the fact that these patients suffered from lung fibrosis, the fibroblast lines isolated from these lungs showed no CCR6 expression. Although based on a small number of cell lines, this lack of expression is remarkable. Even though these diseases are also associated with increased CCL18 levels [25,45], a clear correlation of CCL18 levels with the natural course of the disease has not been shown. In contrast, in IPF, CCL18 is a prognostic marker, and phases of acute exacerbations are accompanied by increasing CCL18 levels [28]. Thus, we speculate that the interaction of CCR6 and CCL18 might be pivotal for the progress of IPF, whereas in fibrosis, due to sarcoidosis or HP, these processes are obviously driven by other factors.

As controls, we used fibroblast lines established from lungs of patients with squamous carcinoma. The tissue was harvested far from the tumour and was therefore considered to be normal. It is of interest that in sera from patients with squamous carcinoma, CCL18 levels are also increased; however, theses CCL18 levels have no impact on the survival time of these patients [32].

We demonstrate here that CCL18-induced profibrotic effects are dependent on the presence of CCR6 and can be blocked by an anti-CCR6 antibody. Although, meanwhile, three further potential CCL18 receptors have been described (GPER [43], PITPNM3 [42] and CCR8 [41]), our data demonstrate an indispensable role for CCR6 in CCL18 signalling. As in our blocking experiments, inhibition by CCR6 is almost complete; a critical role of other receptors in CCL18-triggered activation of lung fibroblasts is rather unlikely. In addition, using flow cytometry, we found no or only marginal expression of GPER, CCR8 or PITPNM3 on our fibroblast lines irrespective of their origin. Both NIH 3T3 cell clones expressed high levels of CCR8, although they responded differently to CCL18 stimulation arguing again for a role of this receptor in matrix induction by CCL18. Interestingly, GPER was expressed only on the surface of the CCR6-negative clone but not on the CCR6-positive clone. Again, as we found CCL18-induced matrix production only in the CCR6^+^/GPER^−^ clone, we conclude that also GPER expression has no role in the CCL18-induced up-regulation of collagen or αSMA.

We would like to mention some limitations of our study. The tissue was harvested more or less randomly without histological or other guidance. Thus, as mentioned above, differences in CCR6 expression might be a result of spatial distribution of positive and negative fibroblasts in the fibrotic lung. In addition, the clinical history of the patients. i.e., limited disease rather at the beginning of the disease versus end-stage lung at lung transplantation may contribute to the differences in CCR6 expression. To clarify this, higher numbers of tissue sections preferably taken from different and defined areas of the lung are necessary. Our results are based on a limited number of fibroblast lines. Additional experiments elucidating the role of CCR6/CCL18 interaction are necessary. Besides fibroblasts, our results indicate that also other cell types in the fibrotic lung express CCR6, although again based on a limited number of analyses. We disregarded these cells here and concentrated on lung fibroblasts. Thus, the contribution of CCR6/CCL18-mediated activation of these cells to the overall fibrotic changes need to be evaluated.

## 5. Conclusions

In summary, we conclude that CCR6 is an important receptor involved in pro-fibrotic activities of CCL18 and that blocking of CCL18/CCR6 interaction or the signalling cascade induced by this interaction is an interesting therapeutic option for IPF.

## 6. Patents

JMQ and GZ hold patents on peptides blocking CCR6 or CCL18 to prevent binding of CCL18 to the receptor.

## Figures and Tables

**Figure 1 cells-13-00238-f001:**
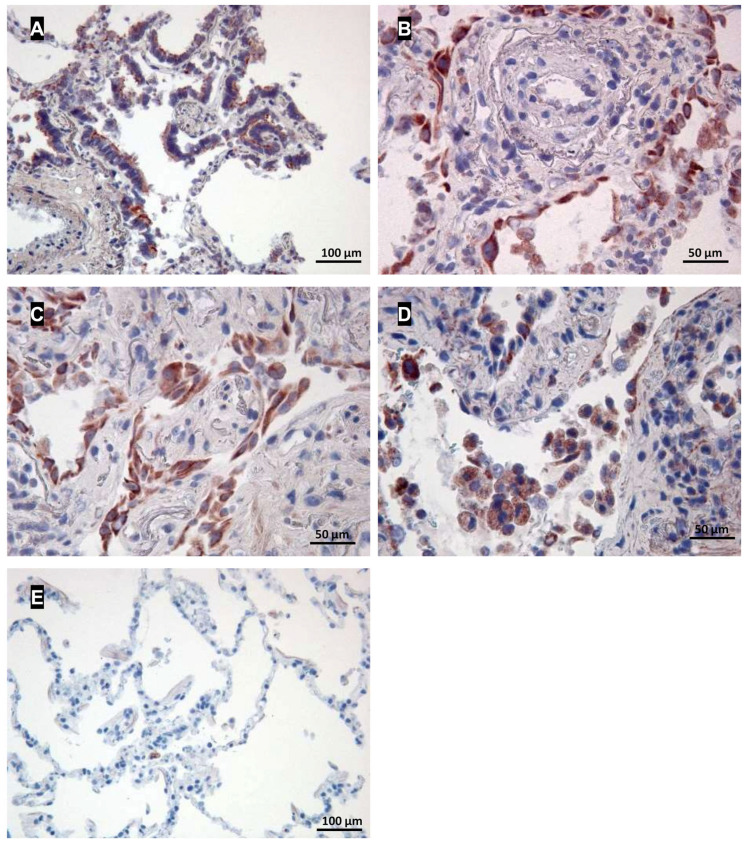
Expression of CCR6 in human lung tissue and lung fibroblast lines. (**A**–**E**): Immunohistochemical staining of CCR6. CCR6 expression (red) is visible in bronchial epithelia (**A**), hyperplastic type II alveolar epithelial cells (**B**), fibroblasts (**C**) and probably macrophages (**D**) in a lung section from an IPF patient. In contrast, in a tumour-free lung sample from a cancer patient without COPD, only a few cells, possibly macrophages, were found to express CCR6 (**E**). Magnification, (**A**–**E**): 200×.

**Figure 2 cells-13-00238-f002:**
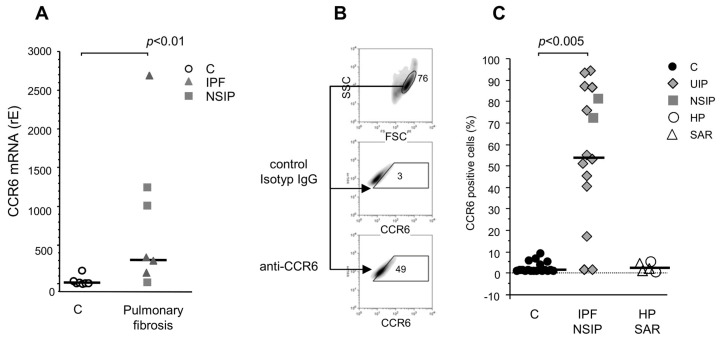
CCR6 mRNA and protein expression in human primary fibroblast lines. (**A**) Quantitative PCR analysis of CCR6 expression in different fibroblast lines from patients suffering from squamous carcinoma (control; n = 7) and pulmonary fibrosis (UIP: n = 4, NSIP: n = 3). Control fibroblasts did not disclose remarkable CCR6 mRNA expression. In contrast, CCR6 mRNA expression is significantly higher in fibroblast lines from pulmonary fibrosis patients (*p* < 0.01). (**B**) Scheme of the flow cytometry strategy: cells are plotted in the forward/sideward scatter (SSC/FSC), and the population containing living cells was gated. Within this gate, a second gate was defined using an isotype control IgG (mid panel) to define the gate for the following CCR6 staining. Cells within this gate are regarded as positive (lower panel). (**C**) Analysis of the flow cytometry revealed a clear up-regulation of CCR6 in fibroblast lines established from pulmonary fibrosis patients (IPF and NSIP) compared to our controls (*p* < 0.005, values of *p* adjusted by Bonferroni–Holm correction) but not in patients suffering from fibrosis due to sarcoidosis (SAR) or hypersensitivity pneumonitis (HP).

**Figure 3 cells-13-00238-f003:**
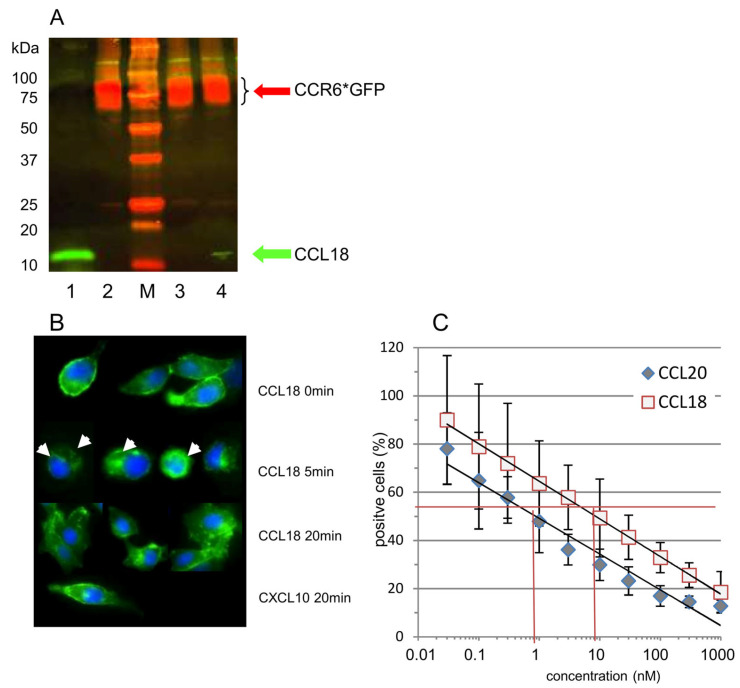
(**A**) Co-immunoprecipitation of CCR6*GFP (red) and CCL18 (green). RLE-6TN cells were loaded with 500 ng/mL recombinant CCL18, and the receptor/ligand complex was gently isolated from the membranes using anti-GFP beads. Lane 1: CCL18 (20 ng); lane 2: lysed RLE-6TN cells only; M: marker; lane 3: CoIP after 10 min of CCL18 incubation; and lane 4: CoIP after 30 min of incubation. CCR6*GFP is visible in all RLE-6TN preparations irrespective of the addition of CCL18 (lanes 2, 3 and 4). CCL18 loaded directly on the gel forms a bright band at the lower end of lane 1 (green). CoIP after 10 min of CCL18 incubation did not reveal a visible CCL18 band. However, after 30 min of incubation, CoIP resulted in a small but visible CCL18 band (lane 4 bottom). The figure depicts one gel out of two independent experiments. (**B**) CCL18 induces receptor internalization in CCR6*GFP-transfected RLE-6TN cells. At time point 0, the receptor is located at the outer rim of the cells. Incubation with rhCCL18 (10 ng/mL) for 5 min induces receptor internalization visible as a ring or as bright dots within the cells (white arrows). After 20 min of incubation with rhCCL18, the CCR6*GFP distribution is merely diffuse. In contrast, incubation with the irrelevant chemokine CXCL10 for 20 min ((**B**), last panel) does not induce receptor internalization (magnification 400×; one of three replicates). (**C**) Loading CCR6 + U937 cells with a wide concentration range of CCL18 or CCL20 revealed a clear dose-dependent internalization of CCR6. It is obvious that for the same effect, approx. 10-fold more CCL18 is necessary compared with CCL20. This figure shows one illustrative series out of three.

**Figure 4 cells-13-00238-f004:**
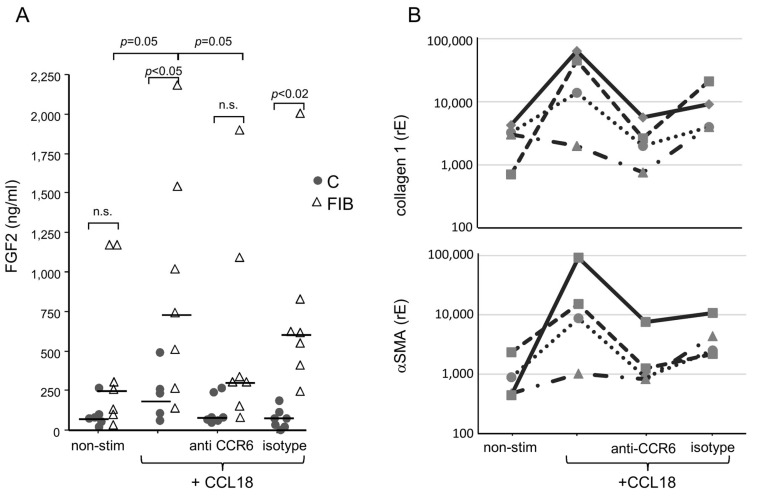
(**A**) FGF2 release of human lung fibroblasts. Fibroblasts from fibrotic tissue (dark bars) release significantly higher levels of FGF2 compared with control fibroblasts (light bars). CCL18 significantly increases FGF2 release in fibrotic fibroblasts but barely in control fibroblasts. The CCL18-induced up-regulation of FGF2 release is abrogated by the addition of an anti-CCR6 antibody. The isotype control induced a nonsignificant reduction in the FGF2 release (C = control: n = 6; Fib = fibrosis: n = 6). (**B**) CCL18-induced upregulation of collagen 1 (upper panel) and alpha-smooth muscle actin (αSMA, (lower panel)) mRNA expression by four different human lung fibroblast lines (indicated by different line patterns) is blocked by anti-CCR6 (rE = relative Expression; n.s. = not significant).

**Figure 5 cells-13-00238-f005:**
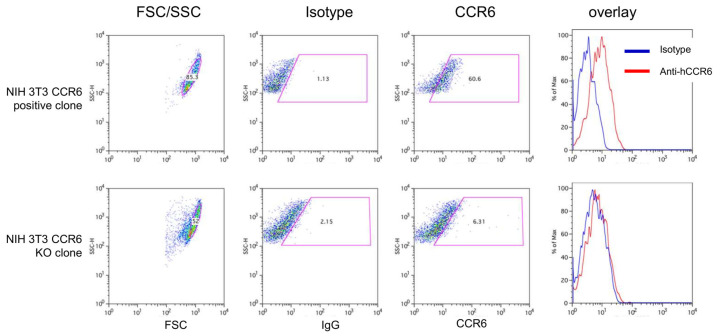
CCR6-positive and CCR6-KO NIH 3T3 cells were harvested and stained using isotype and anti-CCR6 antibodies as described. Gating as well as overlay of the flow cytometric analyses demonstrated that wild-type NIH 3T3 cells clearly express CCR6 on the surface (upper panel). In contrast, the 3T3 CCR6-KO clone does not express CCR6 anymore (lower panel). The figure shows density plot (red colour indicates highest density) and the overlay of isotype and anti-CCR6 staining. Gates (purple boxes) were defined using the isotype control.

**Figure 6 cells-13-00238-f006:**
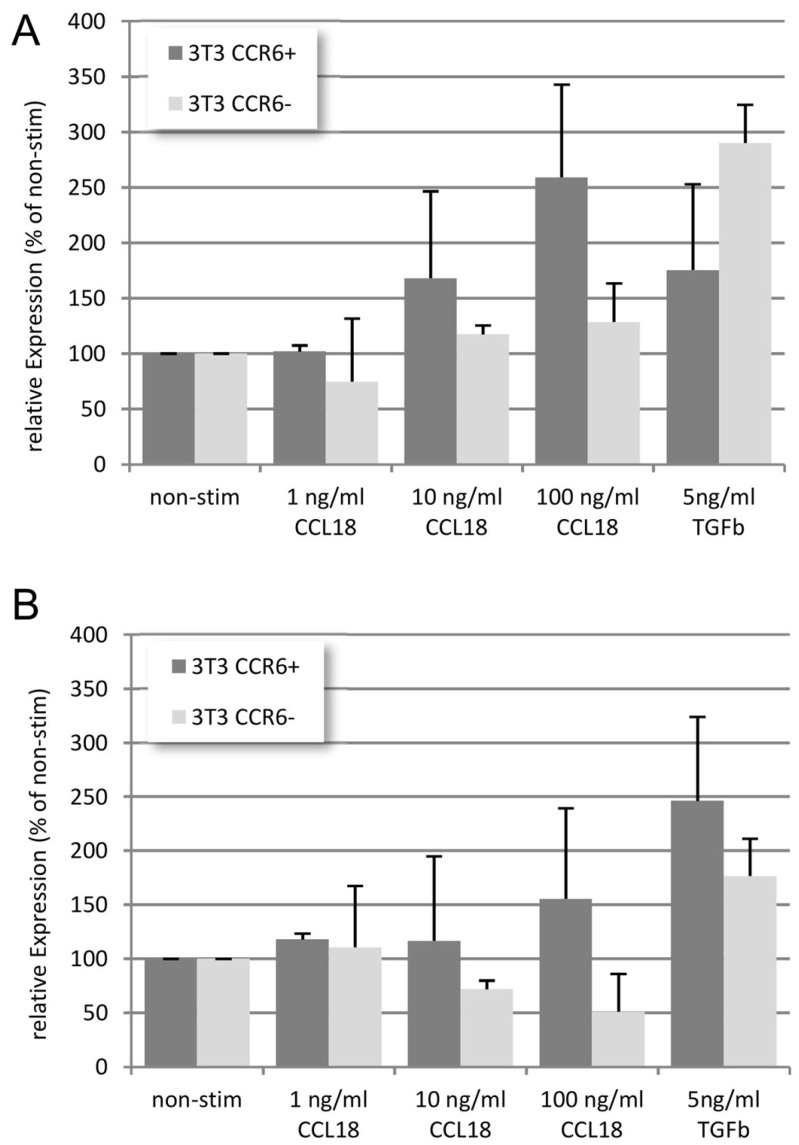
In wild-type NIH 3T3 cells, CCL18 induces a dose-dependent increase in Col1A (**A**) and αSMA (**B**) mRNA expression. TGFβ induced less Col1A but more αSMA mRNA expression compared with the highest dose of CCL18. In contrast, in the CCR6-KO clone, no up-regulation of Col1A mRNA expression was seen. Interestingly, αSMA mRNA expression is down-regulated in this clone (rE = relative Expression; n = 4).

**Table 1 cells-13-00238-t001:** List of primers used in qPCR together with the accession numbers used for generating the primers.

Gene	Accession Number	Forward Primer	Reverse Primer
hGAPDH	NM_002046	CACCAGGGCTGCTTTTAACT	GATCTCGCTCCTGGAAGATG
hCCR6	NM_004367 NM_031409	GCACAAAATGATGGCAGTGG	CCGAAGCACTTCCAGGTTGT
hCollagen 1A1	NM_000088	CCCTGTCTGCTTCCTGTAAACT	CATGTTCGGTTGGTCAAAGATA
hαSMA	NM_001141945	CATCATGCGTCTGGATCTGG	GGACAATCTCACGCTCAGCA
mGAPDH	NM_001289726	GCGAGACCCCACTAACATCAAA	CTTTTGGCTCCACCCTTCAAGT
mCollagen 1A1	NM_007742	TGCTGGGAAACATGGAAACCGA	AGGTTCTCCTTTGTCACCTCGGAT
mαSMA	NM_007392	CACCCAGCACCATGAAGATCAAGA	CCTGTTTGCTGATCCACATCTGCT

## Data Availability

The data presented in this study are available on request from the corresponding author.

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
