# Peer review of "Pro-Fibrotic Effects of CCL18 on Human Lung Fibroblasts Are Mediated via CCR6"

_cells, 2024, doi:10.3390/cells13030238_

Round 1
Reviewer 1 Report
Comments and Suggestions for Authors
1. How about CCR8? Can you prove that CCR8 is not involved in IPF?
2. Fig. 1: To prove which cells express CCR6, fluorescent multiple staining should be performed.
3. Fig. 1: An overall view of the lungs at weak magnification is also needed.
4. Fig. 3: What happens in the dot diagram regarding the FACS analysis? Is it divided into several cell populations?
5. Fig.6: Is FGF2 the only growth factor analyzed? What was the expression of other growth factors that contribute to collagen accumulation?
6. What is the role of other cells expressing CCR6 (bronchial epithelia, hyperplastic alveolar epithelial cells type II, macrophages)?
Author Response
Comments and Suggestions for Authors
We would like to thank the Reviewer for her/his comments on our work.
- How about CCR8? Can you prove that CCR8 is not involved in IPF?
We agree with the reviewer that all putative CCL18 receptors should be mentioned and discussed. We found no CCR8 or PITPNM3 and only marginal GPER (below 10% of the cells) expression on our primary human fibroblasts, irrespective of their origin. In contrast, our NIH 3T3 cells expressed high levels of CCR8. This is not changed by the CCR6-knock down thus, both wt and CCR6 KO cells express CCR8. As there is a striking difference between wt and KO cells, we do not assume that CCR8 plays a major role in this respect.
Cell membrane bound expression of GPER is highly variable in our NIH 3T3 clones. The CCR6- clone is positive for GPER whereas our CCR6+ clone is negative for GPER. Thus, the GPER expression pattern is invers to the CCL18 reactivity. These data suggest that GPER is not linked to the increase of collagen and aSMA in fibroblasts.
Regarding the results of the primary cells, we added in the results section the following sentence:
“Expression of other putative CCL18 receptors like CCR8 [44] or PITPNM3 [45] were not detected. GPER was either not or only marginally detected (<10% of the cells; data not shown).”
In addition, regarding the 3T3 cells the following:
“We would like to mention that GPER was not expressed on the CCR6 positive clone but was present on the CCR6 KO clone (Fig. S7). CCR8 was highly expressed on both clones (Fig. S7). PITPNM3 could not be detected on the clones used here (data not shown).”
We also added one paragraph in the Discussion:
“We demonstrate here that CCL18-induced profibrotic effects are dependent on the presence of CCR6 and can be blocked by an anti-CCR6 antibody. Although meanwhile three further potential CCL18 receptors have been described (GPER [59], PITPNM3 [45] and CCR8 [44]), our data demonstrate an indispensable role for CCR6 in CCL18 signalling. As in our blocking experiments, inhibition by CCR6 is almost complete; a critical role of other receptors in the CCL18 triggered activation lung fibroblasts is ra-ther unlikely. In addition, using flow cytometry, we found no or only marginal expres-sion of GPER, CCR8, or PITPNM3 on our fibroblast lines irrespective of their origin. Both NIH 3T3 cell clones expressed high levels of CCR8 although they respond differ-ent to CCL18 stimulation arguing agains a role of this receptor in matrix induction by CCL18. Interestingly, GPER was expressed only on the surface of the CCR6-negative clone but not on the CCR6 positive clone. Again, as we found CCL18-induced matrix production only in the CCR6+/GPER- clone we conclude that also GPER expression has no role in the CCL18-induced upregulation of collagen or aSMA.”
- Fig. 1: To prove which cells express CCR6, fluorescent multiple staining should be performed.
We agree with the reviewer that this part of our study is not well elaborated. The primary goal of our study was the identification of the receptor responsible for the pro-fibrotic activity of CCL18. As we identified CCR6 as CCL18 receptor, we wonder whether CCR6 is expressed in general in the lung. Fig.1 should demonstrate this. A quantitative or detailed analysis of the expressing cells was not our goal.
- Fig. 1: An overall view of the lungs at weak magnification is also needed.
We apologize but due to time constraints, we could not perform new analyses. A rough overview is seen in lower magnification is seen in Fig. 1A.
- Fig. 3: What happens in the dot diagram regarding the FACS analysis? Is it divided into several cell populations?
We apologize that our figure is not well explained. The figure should just explain the flow cytometric analysis. We reorganized the figure and changed the legend for more clarity.
- Fig.6: Is FGF2 the only growth factor analyzed? What was the expression of other growth factors that contribute to collagen accumulation?
We thank the reviewer for this question. We selected FGF2 as a marker of activation, as it is an important growth factor released by fibroblasts and is released upon fibroblast activation of these cells. Thus, we did not measure other mediators; instead, we concentrated on the direct measurement of the fibrosis-related proteins collagen and aSMA. The question whether CCL18 induces profibrotic mediators which in turn drive the increase in collagen and aSMA is not addressed here.
- What is the role of other cells expressing CCR6 (bronchial epithelia, hyperplastic alveolar epithelial cells type II, macrophages)?
We focus here on the production of collagen and aSMA, which are preferentially produced by fibroblasts. We agree that CCR6 expressing alveolar and epithelial cells are also interesting cells to be analysed. We already have shown that CCL18 induces EMT in cancer cells and change their behaviour into a more aggressive cell (Plönes et al., 2013, PLoS ONE, 8:e53068). Thus, we assume that CCL18 induces EMT in epithelial cells possibly via CCR6, which is under current investigation in our lab.
Currently we do not have data on the role of CCR6/CCL18 interaction in macrophages. However, it is reported that CCL18 induces M2 macrophages from monocytes (Schraufstatter et al., 2012, Immunology, 135:287-298). It is not clear whether CCR6 is involved in this activation but should be tested in the future.
Reviewer 2 Report
Comments and Suggestions for Authors
In this article by Kerstin Höhne and colleagues aimed to identify the receptor for CCL18, a molecule implicated in the progression of idiopathic pulmonary fibrosis (IPF). Using phage display library screening, CCR6 (CD196) was identified as the receptor for CCL18. Expression of CCR6 was found in fibrotic lung tissues and fibroblast lines from affected lungs, with minimal presence in control tissues. Blocking CCR6 reduced CCL18-induced fibrotic responses, suggesting CCR6 as a potential target for IPF treatment. The article is overall well-written, the methods have been properly described, and the experimental designs are appropriate for the conclusions drawn. However, certain things can be improved, and a few questions that I would like to get answered.
Major points:
- In the statistical analysis section, the U-Man Whitney test seemed to be used for everything, with the high N of some of the experiments I wonder if the authors checked normality before applying a non-parametric test that is going to be less powered. Also, have the authors performed any multiple comparison corrections?
- All plots should show the individual replicates inside the boxes, I congratulate the authors on the use of box plots instead of bar plots, but if they can show the individual replicates, we will be able to assess the variability (that is always present in biological samples from different donors) properly.
- How was the cell viability when the flow cytometry was performed? Did the authors use any viability dye such as Hoechst? The expression of CCR6 could be dependent on cell state or viability.
- Please add the scale bar in Figure 1 and for any other microscopy image. I would also suggest quantifying the CCR6 staining if possible.
- Figure 2 could be mixed in a panel with Figure 3.
- The authors mention that samples extracted by TBB show lower CCR6 levels, could the authors elaborate? Is it known if there is some kind of gradient in the lung? Can the authors compare samples from the same patient but at different regions? Lung cells are in general heterogenous, exploring this heterogeneity could be interesting in the future.
- Figure 4 could be mixed with Figure 5 as it is not strong enough as a stand-alone figure.
- Figure 5 has very interesting data, but I would suggest improving the quality of 5A, I would especially encourage using confocal microscopy and a membrane marker (even a simple one like a WGA staining) to show properly that there is internalization. A co-localization with an endosomal marker could also be good but not compulsory.
- IPF fibroblasts seem to be predisposed to the release of FGF2, and CCL18 seems to exacerbate it, can the authors discuss the finding a bit more?
- Figure 6 needs to show all data points so that we can identify the cell line after the treatment to see how much it changes. Also, the isotype for CCR6 seems to have the same effect.
- Figure 7 labels need to be improved, make it clear that CCL18 is being used in the three bars in the middle, and TGFB in the last one. Call the control for its name and show the data points.
- Lines 488-89 speak about data not shown that I would encourage the authors to show.
- I would suggest the authors add a paragraph stating the limitations of their study by the end of the discussion.
- The discussion is written two times with the same words side by side almost, I would suggest deleting the paragraph from the discussion and just leaving the conclusion section.
Minor points:
- The paragraph between lines 87 and 89 is more appropriate for a discussion.
- Lines 311-12 do the authors I suppose the authors refer to Figure 3A.
- Lines 424-31 speak about Figure 7 but call it Figure 8.
- The Discussion is way longer than needed, summarizing it a bit would be helpful.
Author Response
In this article by Kerstin Höhne and colleagues aimed to identify the receptor for CCL18, a molecule implicated in the progression of idiopathic pulmonary fibrosis (IPF). Using phage display library screening, CCR6 (CD196) was identified as the receptor for CCL18. Expression of CCR6 was found in fibrotic lung tissues and fibroblast lines from affected lungs, with minimal presence in control tissues. Blocking CCR6 reduced CCL18-induced fibrotic responses, suggesting CCR6 as a potential target for IPF treatment. The article is overall well-written, the methods have been properly described, and the experimental designs are appropriate for the conclusions drawn. However, certain things can be improved, and a few questions that I would like to get answered.
We thank the reviewer for her/his appreciation.
Major points:
- In the statistical analysis section, the U-Man Whitney test seemed to be used for everything, with the high N of some of the experiments I wonder if the authors checked normality before applying a non-parametric test that is going to be less powered. Also, have the authors performed any multiple comparison corrections?
We thank the reviewer for this hint. The control group contains many zero values and thus, normal distribution cannot be assumed. This is confirmed by the Kolmogorow-Smirnow-Test with Lilliefors-correction. Neither the values of the control group nor of the fibrosis group disclosed normal distribution, and thus non-parametric tests are recommended. In all other tests, the sample size is too small for parametric tests.
We also thank the reviewer for his hint for multiple testing and agree that it should be corrected. Nevertheless, even after Bonferroni-Holm-correction the difference between the controls and fibrosis patients remains significant. We changed the p-value in the figure (<0.005, before 0.0001), mentioned the Bonferroni-Holm-correction in the legend. This also mentioned in the statistics section.
We therefore changed our statistics section:
“As the values of our data do not follow a Gaussian distribution (tested by Kolmogor-ow-Smirnow-Test with Lilliefors-correction) we used the non-parametrical Mann-Whitney test for comparison of independent samples. and Wilcoxon Signed Rank Test for connected data. In cases of multiple testing, the data were first tested for general significant differences using Kruskal-Wallis (independent data) or Friedman Statistics (matched data). In the single comparisons of the individual groups, the val-ues of p are adjusted by Bonferroni-Holm-correction using a web-based calculator [42]. All other tests were performed using StatView for Windows (SAS Institute, Cary, NC, USA). Values of p≤0.05 were considered statistically significant.”
- All plots should show the individual replicates inside the boxes, I congratulate the authors on the use of box plots instead of bar plots, but if they can show the individual replicates, we will be able to assess the variability (that is always present in biological samples from different donors) properly.
We thank the reviewer for this comment and agree. We replaced the box plots in the former figure 6 by individual dots and amended the figure with lines indicating the median. Dots within the bar charts is in many instances (mainly the controls) looks messy.
- How was the cell viability when the flow cytometry was performed? Did the authors use any viability dye such as Hoechst? The expression of CCR6 could be dependent on cell state or viability.
We agree with the reviewer that cell viability is an issue in the flow-cytometric analysis of CCR6 expression. As we mentioned in the Methods section, CCR6 is trypsin-sensitive so that after cell detachment expressed CCR6 may be destroyed. For proper estimation of CCR6 expression, the supplier of the antibody recommends a recovery time of at least 3h, however, the cells must be kept in polypropylene tubes to avoid attachment. If this recovery time is too long, however, we found a substantial increase in dead cells possibly induced by anoikis. Thus, we optimized the recovery time to 4 h to allow clear restoration of CCR6 but to avoid anoikis. Within this time-period cell death is not a major problem. We added “…but not more than 4h” in the methods section to be more specific.
- Please add the scale bar in Figure 1 and for any other microscopy image. I would also suggest quantifying the CCR6 staining if possible.
We apologize for this failt and added scale bars to the pictures in Fig. 1.
- Figure 2 could be mixed in a panel with Figure 3.
We agree with the reviewer and unified figure 2 and 3.
- The authors mention that samples extracted by TBB show lower CCR6 levels, could the authors elaborate? Is it known if there is some kind of gradient in the lung? Can the authors compare samples from the same patient but at different regions? Lung cells are in general heterogenous, exploring this heterogeneity could be interesting in the future.
We thank the reviewer for this comment and agree that these are interesting questions. Our manuscript, as it stands here, was designed to identify a receptor mediating the pro-fibrotic activities of CCR6. We agree with the reviewer, that there are several questions open, which are not addressed by our work. As mentioned by the reviewer, we currently have no clue concerning temporal or spatial distribution of the CCR6 expression in the lung. Both patterns may result in this unequal distribution of CCR6 expressing cells as observed here. Moreover, we have no idea what causes this more or less ectopical expression of CCR6. However, to clarify these questions a higher number of well-characterized samples including the clinical history of the patients is required and should be addressed in the future. We will discuss tis in the limitations section as requested by this reviewer
- Figure 4 could be mixed with Figure 5 as it is not strong enough as a stand-alone figure.
We thank the reviewer and agree to unify both figures.
- Figure 5 has very interesting data, but I would suggest improving the quality of 5A, I would especially encourage using confocal microscopy and a membrane marker (even a simple one like a WGA staining) to show properly that there is internalization. A co-localization with an endosomal marker could also be good but not compulsory.
We thank the reviewer for his appreciation and his idea using a membrane marker. However, re-staining would require us to repeat these time-consuming experiments. Due to time-restrictions, such a repeat is not possible.
- IPF fibroblasts seem to be predisposed to the release of FGF2, and CCL18 seems to exacerbate it, can the authors discuss the finding a bit more?
We thank the reviewer for this comment. Indeed, we agree, as this is also our view. The reason for this increased steady-state FGF2 release is unknown but remarkable that this difference still exist after the long time needed to establish these fibroblast cell lines. Currently we do not have any data explaining this difference; however, it might be related to (epigenetic?) changes also causing the expression of CCR6. We will discuss this with caution, as tis is purely speculative.
- Figure 6 needs to show all data points so that we can identify the cell line after the treatment to see how much it changes. Also, the isotype for CCR6 seems to have the same effect.
We agree with the reviewer and redraw the figure to show all data points. In addition, we applied Bonferroni-Holm-correction for multiple comparisons. We again thank the reviewer to bring up this point, as the Bonferroni-Holm-correction even sharpens the message of Fig. 6 (now Fig. 4). CCL18 induces a significant increase in FGF2 release. After Bonferroni-Holm-correction there is a significant difference between the CCL18 induced FGFs release of controls and fibrosis fibroblasts. This significant difference is lost in the anti-CCR6 treated cultures but it remains significantly different in the isotype control treated cultures although there is also a substantial reduction induced by the control antibody. This difference demonstrates the blocking CCR6 is more effective than the overall non-specific reduction by the isotype control. The reason for this non-specific reduction is unclear. The antibodies used are described as azide free and suitable for cell culture. However, we cannot rule out that either the antibody itself or an ingredient of the buffer is responsible for this reduction, as unfortunately we did not dialyze the antibodies.
- Figure 7 labels need to be improved, make it clear that CCL18 is being used in the three bars in the middle, and TGFB in the last one. Call the control for its name and show the data points.
We apologize for the missing label in this figure. This is now corrected.
- Lines 488-89 speak about data not shown that I would encourage the authors to show.
We thank the reviewer for bringing this up, however, here we disagree with the reviewer as our data a very preliminary. We remove this sentence and instead we extend the discussion on difficulties using a human cytokine (which does normally not exist in this organism) in a mouse model of a human disease. First, the bleomycin model is based on a strong inflammatory reaction in the initialization phase followed by a subsequent and limited fibrosis. Currently, as questioned above, we have no idea how CCL18 influences the inflammatory and/or fibrotic processes in this model. In addition, we have no information (or only limited) on the receptor expression in this model. This has to be clarified in the future and better model should be developed.
- I would suggest the authors add a paragraph stating the limitations of their study by the end of the discussion.
We agree and ad a description of the limitations of the study.
- The discussion is written two times with the same words side by side almost, I would suggest deleting the paragraph from the discussion and just leaving the conclusion section.
We thank the reviewer and agree to remove the last paragraph. We apologize as it is a remnant of a former version and should not be there.
Minor points:
- The paragraph between lines 87 and 89 is more appropriate for a discussion.
We thank the reviewer for this comment. This sentence is part of the objectives of our manuscript, we therefore think that we should keep this sentence. However, we agree with the reviewer that this is not clear as this and the following paragraph stands right now. We deleted the first two sentences of the following paragraph and unified both paragraphs to make our rational more clear.
- Lines 311-12 do the authors I suppose the authors refer to Figure 3A.
Most of the figures are redrawn and relabeled and the assignment is updated.
- Lines 424-31 speak about Figure 7 but call it Figure 8.
We apologize for this fault. Indeed, the figures were interchanged.
- The Discussion is way longer than needed, summarizing it a bit would be helpful.
We thank the reviewer and try to condense the discussion. However, as new topics had to be discuss, we fear that we failed in these attempts…..
Round 2
Reviewer 1 Report
Comments and Suggestions for Authors
The authors have addressed the reviewers' questions sincerely. Therefore, we would like to accept the paper.